

# Lipid extraction from microalgae using pure caprolactam-based ionic liquids and with organic co-solvent

Rania A. Naiyl[1,2,3], Fredrick O. Kengara[1,2,4], Kirimi H. Kiriamiti[2,5] and Yousif A. Ragab[3]

[1] Department of Chemistry and Biochemistry, School of Science & Aerospace Studies, Moi University, Eldoret, Uasin Gishu, Kenya
[2] Africa Centre of Excellence II in Phytochemicals Textile and Renewable Energy (ACE II PTRE), Moi University, Eldoret, Uasin Gishu, Kenya
[3] Department of Chemistry & Mohamed Obaid Mubarak (MOM) Laboratories, Faculty of Engineering and Technology, University of Gezira, Wad Medani, Gezira State, Sudan
[4] School of Pure and Applied Sciences, Bomet University College, Bomet, Rift Valley, Kenya
[5] Department of Chemical and Process Engineering, School of Engineering, Moi University, Eldoret, Uasin Gishu, Kenya

Corresponding authors
Rania A. Naiyl, raniaawad@mu.ac.ke
Fredrick O. Kengara,
fkengara@googlemail.com

## ABSTRACT

**Background:** The main process limitation of microalgae biofuel technology is lack of cost-effective and efficient lipid extraction methods. Thus, the aim of this study was to investigate the effectiveness and efficiency of six caprolactam-based ionic liquids (CPILs) namely, Caprolactamium chloride, Caprolactamium methyl sulphonate, Caprolactamium trifluoromethane sulfonate, Caprolactamium acetate, Caprolactamium hydrogen sulphate and Caprolactamium trifluoromethane-acetate—for extraction of lipids from wet and dry *Spirulina platensis* microalgae biomass. Of these, the first three are novel CPILs.

**Methods:** The caprolactam-based ionic liquids (CPILs) were formed by a combination of caprolactam with different organic and inorganic Brønsted acids, and used for lipid extraction from wet and dry *S. platensis* microalgae biomass. Extraction of microalgae was performed in a reflux at 95 °C for 2 h using pure CPILs and mixtures of CPIL with methanol (as co-solvent) in a ratio of 1:1 (w/w). The microalgae biomass was mixed with the ILs/ methanol in a ratio of 1:19 (w/w) under magnetic stirring.

**Results:** The yield by control experiment from dry and wet biomass was found to be 9.5% and 4.1%, respectively. A lipid recovery of 10% from dry biomass was recorded with both caprolactamium acetate (CPAA) and caprolactamium trifluoroacetate (CPTFA), followed by caprolactamium chloride (CPHA, 9.3 ± 0.1%). When the CPILs were mixed with methanol, observable lipids' yield enhancement of 14% and 8% (CPAA), 13% and 5% (CPTFA), and 11% and 6% (CPHA) were recorded from dry and wet biomass, respectively. The fatty acid composition showed that $C_{16}$ and $C_{18}$ were dominant, and this is comparable to results obtained from the traditional solvent (methanol-hexane) extraction method. The lower level of pigments in the lipids extracted with CPHA and CPTFA is one of the advantages of using CPILs because they lower the cost of biodiesel production by reducing the purification steps.

**Conclusion:** In conclusion, the three CPILs, CPAA, CPHA and CPTFA can be considered as promising green solvents in terms of energy and cost saving in the lipid extraction and thus biodiesel production process.

## INTRODUCTION

Biodiesel is a clean and renewable energy source that is considered as an important option to petroleum consumption (*Gonçalves, Pires & Simões, 2013*). Petroleum retails at a high cost, thus threatening energy security, in addition to causing global climate change concerns (*Pragya, Pandey & Sahoo, 2013*). Biodiesel is primarily made from oil obtained from both edible and non-edible plants, and residual waste (*Pragya, Pandey & Sahoo, 2013*). The use of these plants has serious drawbacks, including high costs, food shortages, and a lack of steady and reliable supply. These difficulties could be mitigated by the synthesis of biodiesel from microalgae, that has long been considered as a promising potential alternative biomass for biodiesel production due to its extremely fast biomass productivity rate (*Arumugam et al., 2013*). Other advantages of microalgae include higher lipid accumulation capacity and its requirement for lesser land compared to other biofuel crops.

Nevertheless, the major constraint in the biofuel production from microalgae, is the lack of cost-effective and efficient extraction and transesterification of lipids. Although higher lipid yields have been recorded after pre-treatment of microalgae with various cell disruption methods (*Halim, Danquah & Webley, 2012*) such as bead milling, microwave, and ultrasonication, the additional energy needed makes the process economically unviable. On the other hand, conventional lipid extraction methods also require refluxing with flammable and highly toxic organic solvents. Therefore, the exploration of alternative microalgal lipid processing methods that are simpler, cost-effective, and environmentally friendly, has become increasingly necessary.

Ionic liquids (ILs) have lately been identified as promising green solvents in the extraction of microalgal lipids based on their fascinating physicochemical properties such as being non-volatile, non-flammable, chemically and thermally stable, and having the potential for recovery and design (*Zhao & Baker, 2013*). Moreover, ILs can dissolve essential biopolymers like cellulose and lignin and, as a result, induce the structure disruption of algae cells or affect the permeability of cell walls, depending on their cation and anion structure (*Cevasco & Chiappe, 2014*).

Majority of the research on ILs extraction of lipids from algae has been based on the use of imidazole-based ILs. The high cost of imidazole-based ionic liquids may limit their availability and applicability (*Andreani & Rocha, 2012*; *George et al., 2015*). Thus, the use of protic ionic liquids (PILs) has attracted significant attention as a novel technology for microalgal lipid extraction and biodiesel production (*Kim et al., 2012*; *Kim et al., 2013*;

*Choi et al., 2014*; *Chiappe et al., 2016*). PILs are substantially less expensive than common ILs because they can be synthesized by neutralizing a selected base with a protic acid under mild conditions (*Greaves & Drummond, 2008*; *Hayes, Warr & Atkin, 2015*; *Xu & Angell, 2003*). Besides, PILs are less toxic (*Oliveira et al., 2016*; *Shankar et al., 2019*; *Bodo et al., 2021*), and are known to form strong hydrogen bonds due to their labile protons (*Chhotaray, Jella & Gardas, 2014*). In particular, caprolactam-based ionic liquids (CPILs) have recently been identified as lipid extraction solvents (*Shankar et al., 2019*) and catalysts for lipid transesterification reactions (*Luo et al., 2017*). The findings showed the capability of these CPILs to disrupt cells and extract lipids in a single step.

Despite their many potential advantages, CPILs are rarely synthesized and their application is therefore limited. However, CPILs researchers are currently working to produce new forms of CPILs that could be used as green solvents. In spite of these efforts, the efficacy of CPILs extraction of lipids from *S. platensis* is hardly reported.

To establish whether it is possible to improve the long-term viability and sustainability of the extraction procedure for lipids, we have investigated the effectiveness and efficiency of six CPILs—Caprolactamium chloride (CPHA), Caprolactamium methyl sulphonate (CPMS), Caprolactamium trifluoromethane sulfonate (CPTFS), Caprolactamium acetate (CPAA), Caprolactamium hydrogen sulphate (CPSA) and Caprolactamium trifluoromethane-acetate (CPTFA)—for extraction of lipids from wet and dry *S. platensis* microalgae biomass. Of these, the first three are novel ILs (*Naiyl et al., 2021*), whereas the others—except for Caprolactam acetate—were used for the first time in lipids' extraction. Extractions with both pure ionic liquids and ionic liquids/methanol mixtures were done to establish whether organic co-solvents could improve lipid extraction yields.

## MATERIALS AND METHODS

### Sources of microalgae, chemicals and reagents

*Spirulina platensis* dried biomass was obtained from an algae cultivation pond at Masinde Muliro University of Science and Technology (coordinates: 0.5947° N, 34.7803° E), Kenya. Caprolactam (CP 99%), Methane sulphonic acid ($CH_3SO_3H$, 99 %), Trifluoromethanesulphonic acid ($CF_3SO_3H$, 99%) and n-hexane were supplied by Sigma-Aldrich (Darmstadt, Germany). Hydrochloric acid (HCl, 37%), Sulfuric acid ($H_2SO_4$, 98%), Trifluoroacetic acid ($CF_3CO_2H$, 98%) and Methanol ($CH_3OH$ 99.8%) were supplied by Labo Chem PVT (Mumbai, India), (Toluene, 99.6%) was supplied by VWR (Shanghai, China) and Acetic acid ($CH_3COOH$, 99.6%) was obtained from M&B (Birmingham, England).

### Caprolactam ionic liquid synthesis and characterization

The CPILs used for lipid extraction experiments are listed in (Table 1), and were synthesized by adding equimolar quantities of caprolactam and an acid (hydrochloric, methane sulphonic, trifluoromethanesulphonic, acetic, Trifluoroacetic, and sulfuric), followed by stirring at room temperature for 24 h. Four of these ILs are liquids at room temperature, namely CPAA, CPSA, CPTFA and CPTFS. The methodologies for synthesis and characterization are explained in detail elsewhere in the literature (*Naiyl et al., 2021*).

**Table 1 Caprolactam-based ionic liquids used in the study.**

| CPILs | Abbreviation | Structural formula | Water content (w/w %) |
|---|---|---|---|
| Caprolactamium chloride | CPHA | | 0.74 |
| Caprolactamium methanesulphonate | CPMS | | 1.85 |
| Caprolactamium acetate | CPAA | | 4.52 |
| Caprolactamium hydrogen sulphate | CPSA | | 1.21 |
| Caprolactamium trifluoromethane acetate | CPTFA | | 0.65 |
| Caprolactamium trifluoromethane sulphonate | CPTFS | | 1.53 |

## Extraction of lipids using the traditional Hexane: methanol method

As described by *Chiappe et al. (2016)* a hexane-methanol mixture (54:46, v/v, 150 mL) was used to extract lipids from wet (80%) and dry microalgae (3.0 g) by the Soxhlet extraction method, which has three main compartments. 250 mL round bottom flask holding the solvent, extraction chamber and condenser. First, the sample was placed in a porous thimble, the flask was heated, and then the solvent was evaporated and carried to a condenser, where it was converted to a liquid and collected in the extraction chamber containing the sample. As the solvent passed through the sample, the lipids were extracted and transported to the flask. This process lasted 10 h. After extraction, a rotary evaporator was used to remove the solvent. Then extracted lipids fraction was transferred into a weighed beaker and dried in an oven at 60 °C until it reached a constant weight. The experiments were carried out in duplicate and the crude lipids extraction yield was then calculated using the following equation:

$$R_{lipid}\% = \frac{W_{lipid}}{W_{biomass}} \times 100 \tag{1}$$

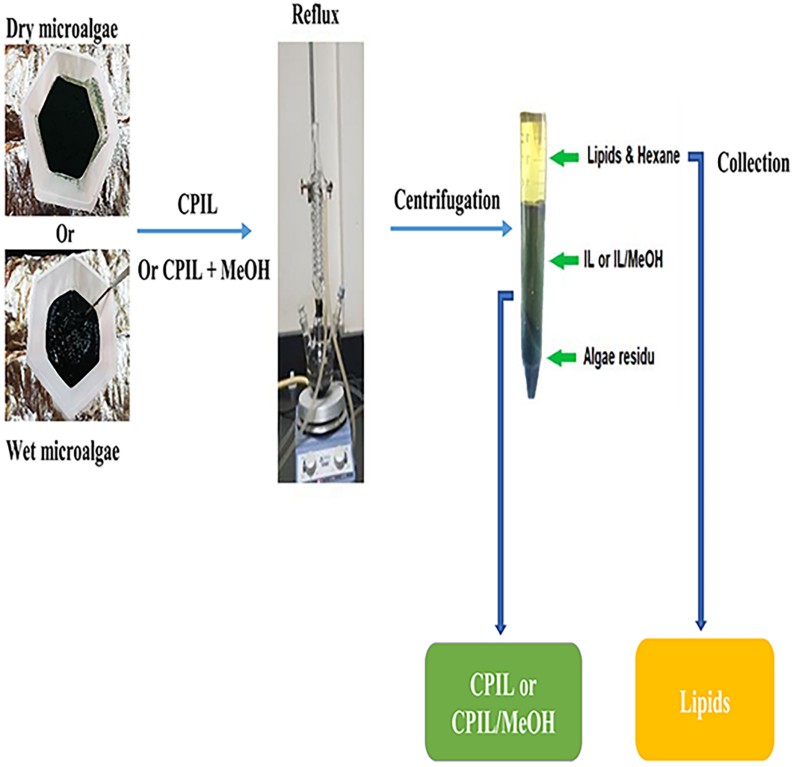

**Figure 1  A schematic presentation of the lipid extraction process.**

where the $R_{lipid}$ and $W_{lipid}$ are the recovery and the weight of crude lipid extracts, respectively. $W_{biomass}$ is the initial dry biomass weight (g).

## Lipid extraction using ionic liquids
### The effect of reaction time and temperature on lipid extraction

Three ionic liquids, CPAA, CPHA, and CPSA were utilized for lipid extraction for 5 h at 75 °C and for 2 h at 95 °C using reflux (150 mL round bottom flask fitted with a condenser). The dry biomass of microalgae was mixed with the ionic liquid in a ratio of 1:19 (w/w) under magnetic stirring. After extraction, a tri-phasic system was obtained by centrifugation. The top phase contained lipids, the middle phase contained IL with methanol, and the bottom phase contained the algae residue. The upper lipid phase could not be easily retrieved due to the small scale of the experiment and therefore, the mixture was treated with n-hexane (10 mL) or a mixture of hexane: methanol 2:1 (v/v) to ascertain the actual lipid yield. The recovered n-hexane phase was washed two times with water to remove polar compounds. The lipid fraction was dried in a thermostat oven at 60 °C until it reached a constant weight, and the residue was weighed to calculate the gravimetric yield using Eq. (1). The overall lipid extraction process is shown in Fig. 1.

## Lipid extraction using pure ionic liquids

Lipid extraction using the six-caprolactam ionic liquids was performed at 95 °C for 2 h. Afterwards, lipids were extracted from the ionic liquids following the same procedure described above. In the case of CPSA, CPMS, and CPHA, hexane: methanol 2:1 (v/v) was used, rather than hexane, because these CPILs solidify after mixing with hexane.
The solidification happened, due to the fact that, CPHA and CPMS are solid at room temperature, whereas CPSA is highly viscous. Thus, when hexane was added at room temperature (in our case this ranged from 18 to 20 °C), and being immiscible with the ILs, there was a decrease in temperature which led to solidification of the mixture. Therefore, methanol was added because it is miscible with the ILs, to ease their transfer to centrifuge tubes.

## The effect of organic co-solvents on ionic liquid extraction of lipids

Lipids were extracted using mixtures of IL and methanol (1:1 w/w) as co-solvent. Methanol (MeOH) was also used separately as a negative control. Dry biomass of microalgae (0.5 g) was mixed with 4.8 g each of ionic liquid and methanol in a round bottom flask (150 cm$^3$) equipped with a condenser.

The extraction was conducted at 95 °C for 2 h under magnetic stirring at 600 rpm. Thereafter, lipids were extracted with hexane (2 × 5 mL). To facilitate the faster separation of layers, the centrifugation of the mixture was further performed at 4,000 rpm for 30 min.

## Extraction of lipids from wet biomass

To find out the effect of water content on lipids extracted from microalgae, 4 g of distilled water were added to 1 g of dry biomass of *S. platensis* to form a wet biomass with 80% water content. The lipids extraction was performed using mixtures of IL/methanol (1:1 w/w) to wet biomass at the ratio of 19:1 at 95 °C for 2 h, following the procedure described earlier. The lipids extraction yield was calculated using Eq. (1).

## Lipid transesterification and FAMEs analysis

Using the method reported by Christie & Han (2012) various amounts of extracted lipids (10–70 mg) were converted to FAMEs-biodiesel *via* acid/catalyzed esterification/ transesterification reaction. Lipids were dissolved in hexane (1 mL) in a stoppered tube, and 2 mL of 1% sulfuric acid in methanol was added. The mixture was left overnight at 50 °C. 5 mL of water containing sodium chloride (5%) was added and the esters were extracted with n-hexane (2 × 5 mL) using a Pasteur pipette, and the layers were separated. The n-hexane layer was washed with water (4 mL) containing potassium bicarbonate (2%) and dried over anhydrous sodium sulphate. The solution was filtered and the solvent was evaporated.

The recovered FAMEs were analyzed using gas chromatograph (GC3420Al MRC, Madrid, Germany) equipped with a flame ionization detector (FID) and an Agilent CPSil 88 capillary column was used to analyze the recovered FAME, using nitrogen as a carrier gas and other gases such as hydrogen and air. The FID and the injector port temperatures were kept at 260 °C and 240 °C, respectively. The injection volume was

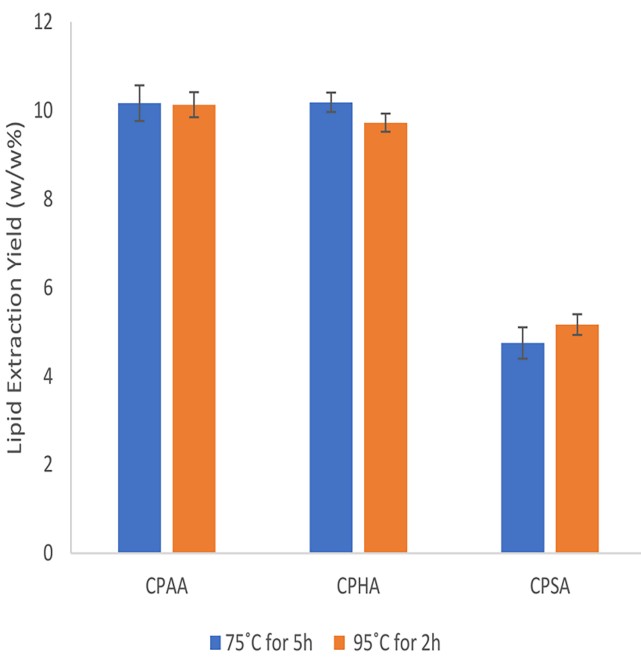

**Figure 2 The yield of extracted Lipids by caprolactam ionic liquids at 75 °C for 5 h and 95 °C for 2 h.**

0.5 μL and gas flow rate was 100 mL/min. The temperature program was held at 150 °C for 1 min, increased to 220 °C at 10 °C/min and held for 2 min, then increased to 240 °C at 3 °C/min, and finally maintained at 240 °C for 8 min. For external calibration, a 37-component FAMEs standard mixture was used.

## Data analysis

The analysis was performed in duplicate and the obtained data was expressed as (Mean ± standard deviation). The $T$-test was used to compare results with controls, and an effect was considered to be significant when $P \leq 0.05$.

## RESULTS AND DISCUSSION

### Optimization of lipid extraction time and temperature

Figure 2 shows the comparison of lipids yield between extraction for 5 h at 75 °C and for 2 h at 95 °C of three selected CPILs. In order to minimize reaction time, long period/low temperature and short period/high temperature were compared. The results showed that there was no significant difference in lipids extraction yields ($P$-value = 0.23) of CPHA, CPAA and CPSA at 75 °C for 5 h and at 95 °C for 2 h. Therefore, a synthetic ionic liquid was used for lipids extraction at 95 °C for 2 h.

### Extraction of lipids by a conventional method and pure ILs

The total recovered lipids obtained by conventional Soxhlet extraction using (Hexane–MeOH) as solvents, as well as the extraction yields of the synthesized CPILs, are shown in Fig. 3. The yield from the conventional organic solvent extraction method was 9.5 ± 0.23%, which is similar to the yield obtained by *Mandal et al. (2013)*, who used
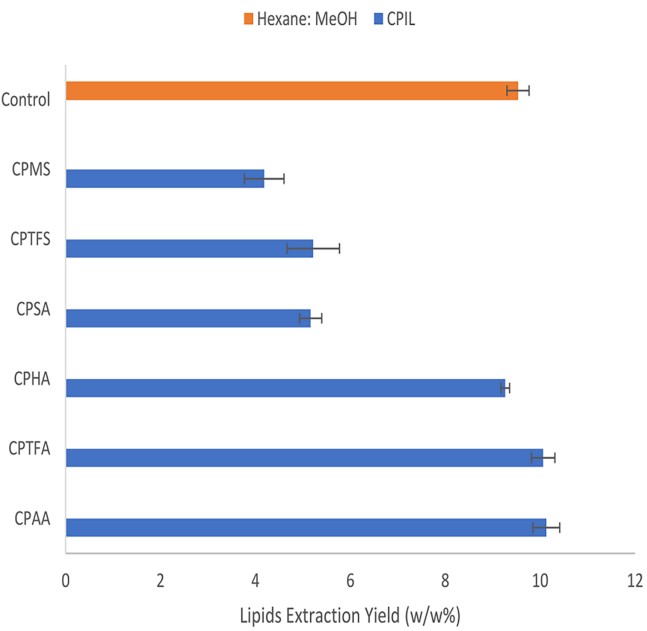

**Figure 3** **The yields of extracted lipid by pure caprolactam ionic liquids and Hexane: MeOH (control).**

the following conditions: Chloroform: methanol (2:1), 6 h at r.t. Three of the CPILs had no significant differences in extraction yields compared to the control experiment ($P > 0.05$). In particular, CPAA, CPTFA and CPHA had lipid yields of (10.1 ± 0.28%, $P$-value = 0.153), (10.1 ± 0.25%, $P$-value =0.159), and (9.3 ± 0.1%, $P$-value = 0.326), respectively. This contrasts with previous findings on lipid extracts from dried and dehydrated marine *Nannochloropsis oculata* and *Chlorella salina* microalgae, which showed that using CPAA for extraction resulted in the lowest yield compared to the control (*Shankar et al., 2019*). The reason for this can be due to the fact that the cell wall structures of microalgae, which contains cellulose, glycoprotein, silica, and peptidoglycan (*Zhou et al., 2019*), may vary from one type to another. Therefore, the ability of different ILs to penetrate the cell wall may also vary. Hence, the wall structure of *S. platensis* might be more affected by CPAA. On the other hand, lower yields (*circa*. 5%) were obtained using CPSA ($P$-value = 0.003), CPTFS ($P$-value = 0.031), and CPMS ($P$-value = 0.01). Overall, the CPILs containing sulphate and sulphonate anions recorded the lowest lipids yield relative to the control experiment.

## The effect of organic co-solvents on ionic liquid extraction of lipids

Figure 4 shows the yields for extraction of lipids from dry biomass using mixtures of ionic liquids and methanol (as co-solvent). Methanol (MeOH) was also used separately as a negative control for comparative purposes and the obtained lipids yield was (1.31 ± 0.27%). The highest yield was (14.2 ± 0.11%, $P$-value = 0.007) and (13.1 ± 0.1%, $P$-value = 0.02) for the CPAA/MeOH and CPHA/MeOH mixtures, respectively. Here, a significant increase ($P < 0.05$) over that of pure CPAA and CPHA was observed. However, the lipids yield of the CPTFA/MeOH mixture (11.1 ± 0.13%, $P$-value = 0.028) is slightly increased

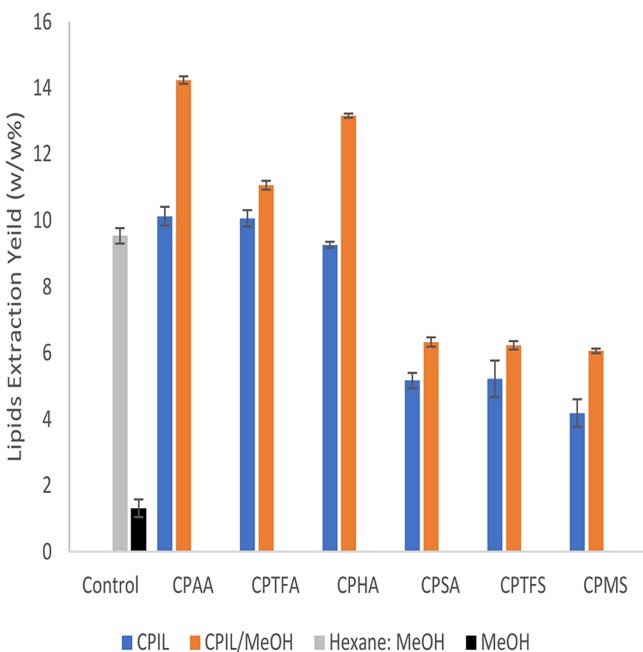

**Figure 4 Comparison of lipid extraction yields by CPILs and mixtures of CPIL/Methanol (1:1).**

compared to the pure one, but was still significantly different from that of the control experiment ($P < 0.05$). This can be attributed to different interaction mechanisms between the ILs and methanol. The mixtures of methanol with CPSA, CPTFA, and CPMS showed an improvement in lipid yield of around (6%).

The co-solvent effect can be explained by enhancement of microalgae cell disruption through the action of the polar methanol, which improves the efficiency of lipid extraction from biomass (*Halim, Danquah & Webley, 2012*; *Dong et al., 2016*). The reason behind this is that some non-polar lipids are found in the cytoplasm as a complex with polar lipids. This complex is strongly bound to proteins in the cell membrane *via* hydrogen bonds. Therefore, the ILs and MeOH, which are hydrophilic in nature, can disrupt lipid-protein associations by forming hydrogen bonds with the polar lipids of the complex (*Halim, Danquah & Webley, 2012*). As a result, whereas ILs improve the permeability of the cell wall, methanol accelerates the precipitation of lipids from the cell (*Zhou et al., 2019*). It is also thought that the action of the ILs—Methanol system creates a more hydrophobic environment, which makes lipid transfer easier (*Zhou et al., 2019*). The same author has also reported that, the addition of methanol may reduce the viscosity of the ILs, boosting the possibility of hydrogen bonds forming between fibres on the microalgae cell wall and ionic liquids.

Moreover, the differences in intra molecular interactions of these ionic liquids seem to be the main reason for their capability to form hydrogen bonds with the microalgae cell walls, and thus lead to the differences in their effectiveness for lipid extraction.

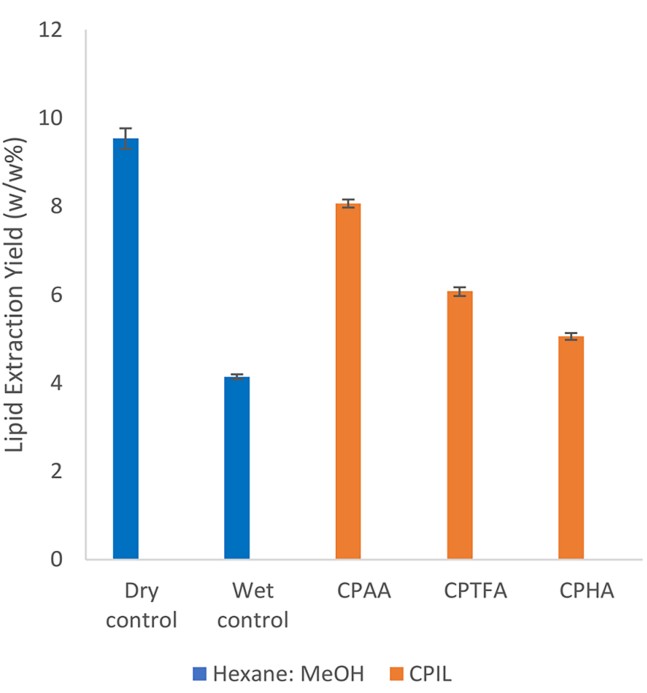

**Figure 5  Lipid extraction yields from wet *Spirulina platensis* biomass.**

## The effect of IL/Methanol mixture on lipid extraction from wet biomass

In order to reduce the cost of extraction of lipids from microalgae, we have also studied the potential of these CPILs to extract the lipids from wet biomass to avoid the drying stage, which is considered a major cause for the high cost of biodiesel production.

For this investigation, we used the three mixtures of CPAA/MeOH, CPTFA/MeOH and CPHA/MeOH, which recorded the highest lipid yields from dry biomass. The control sample of wet *S. platensis* biomass (80%) provided a lipid yield of (4.1 ± 0.06%). The results show that CPAA/MeOH mixture provided the maximum yield of (8.07 ± 0.09%, *P*-value = 0.001), followed by CPTFA/MeOH (6.1 ± 0.1%, *P*-value = 0.005) and CPHA/MeOH mixture recorded the lowest yield of (5.1 ± 0.08%, *P*-value = 0.008). As can be seen the three mixtures provided a higher yield ($P < 0.05$) over that of the control (Fig. 5). On the other hand, the lipids yield of CPAA is almost similar (*P*-value = 0.047) to that of the control sample from dry biomass (9.5 ± 0.23%). Therefore, from an economic point of view, CPAA could be the most promising ionic liquid for production of biodiesel from *S. platensis* biomass—in terms of energy and cost savings, when compared to conventional extraction processes.

## Determination of fatty acids composition in the lipid fraction

The experiments of CPILs lipids extraction with dried microalgae show that CPAA/MeOH, CPHA/MeOH, and CPTFA/MeOH mixtures gave the highest lipid yields. However, it is necessary to compare the nature of these lipids with those obtained from the control sample. Thus, the lipids recovered from the control, CPAA/MeOH, CPHA/MeOH, and CPTFA/MeOH mixtures were trans-esterified and their FAME composition

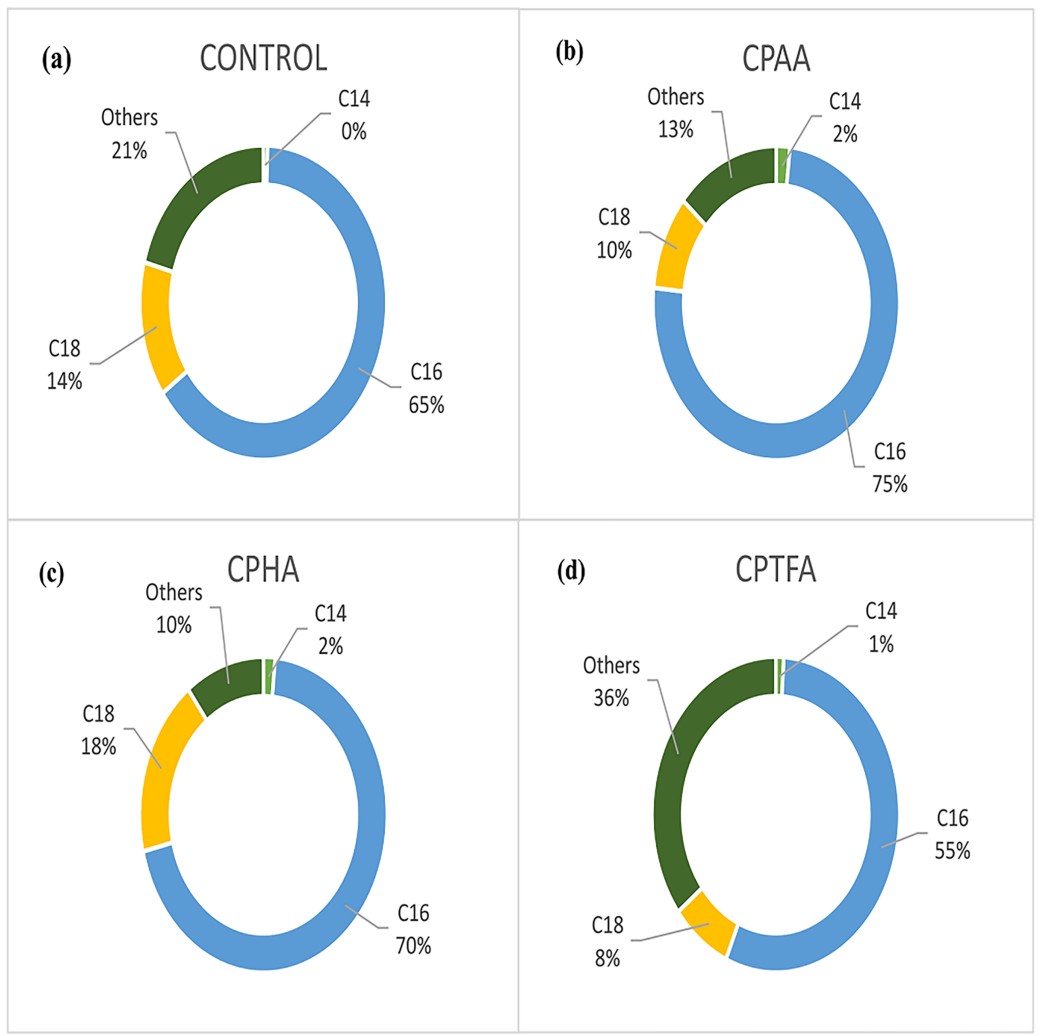

**Figure 6 The relative fatty acids composition in lipids recovered by (A) control (Hexane: MeOH), (B) by CPAA/MeOH, (C) by CPHA/MeOH, and (D) by CPTFA/MeOH mixtures.**

identified using GC-FID analysis. The percentages of FAMEs in the lipid fractions of microalgae are shown in Fig. 6. The fatty acids methyl esters found in *S. platensis* biomass-based biodiesel were myristic (14:0), palmitic (16:0), stearic (18:0), palmitoleic (16:1), oleic (18:1), linoleic (18:2), eicosenic (20:1) and lignoceric (24:0). Among all fatty acid methyl esters, palmitic acid methyl ester (PAME) was the most abundant in all the lipid extracts, followed by oleic and stearic methyl esters. Similar observations were made by *Fattah et al. (2020)* who found FAME compositions obtained by enzymatic synthesis from *Chlorella* sp. and *Spirulina* sp., were mainly predominated by palmitic acid, followed by oleic and stearic methyl esters. The lipid profiles of the CPAA (Fig. 6B) and CPHA (Fig. 6C) extracts produced $C_{16}$ (75% and 70%), respectively, which is higher compared to the control ($C_{16}$, 65%) as shown in Fig. 6A, whereas both CPILs produced $C_{18}$ (10% and 18%, respectively), which were quite similar to the control sample ($C_{18}$, 14%).

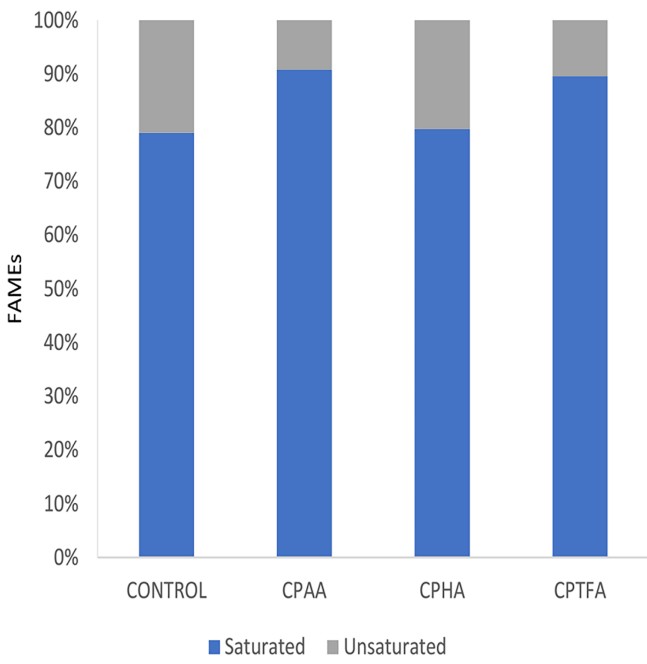

**Figure 7 Saturated and unsaturated fatty acid methyl esters profile in biodiesel produced by CPIL/MeOH mixtures.**

In contrast, the lipid profile of the CPTFA extract (Fig. 6D) produced $C_{16}$ (55%) and $C_{18}$ (8%), which were lower than those produced by the control.

On the other hand, the biodiesel produced by the CPILs had a higher proportion of saturated fatty acids (80–90%) than unsaturated fatty acids (9–21%) as shown in Fig. 7, which could be advantageous because saturated FAMEs have superior burning qualities while unsaturated FAMEs have better fluidity in cold temperatures (*Knothe, 2005*). Long-chain saturated fatty acids like palmitic and stearic acid are quite desirable (*Hayyan et al., 2011*) due to the fact that biodiesel with a high saturated fatty acid content has a higher oxidation resistance (*Mostafa & El-Gendy, 2013*).

Overall, the lipid profiles and yields obtained indicate that the CPIL solvents, CPAA, CPHA, and CPTFA, are better than or as effective as the conventional organic solvents in the extraction of lipids from microalgae, and can thus be used as alternative green solvents. Moreover, the CPILs can be separated as discrete phases at the end of the process, which means that they can be recycled for more extractions.

## The influence of CPILs on the color of the lipid fractions

The undesirable presence of lipid-soluble carotenoids and chlorophyll is responsible for the green color of plant-derived lipids (*Shankar et al., 2019*). The high intensity of the colour would increase the number and length of biodiesel production processes, resulting in less sustainable economics. Figs. 8A–8C shows equal lipids fractions (w/w, diluted in hexane to facilitate color observation) produced by CPAA/Me, CPHA/Me and CPTFA/Me mixtures, respectively. The color intensity of the lipids extracted by CPAA in Fig. 8A seems to be much darker than the lipids extracted by CPHA and CPTFA in Figs. 8B and 8C,

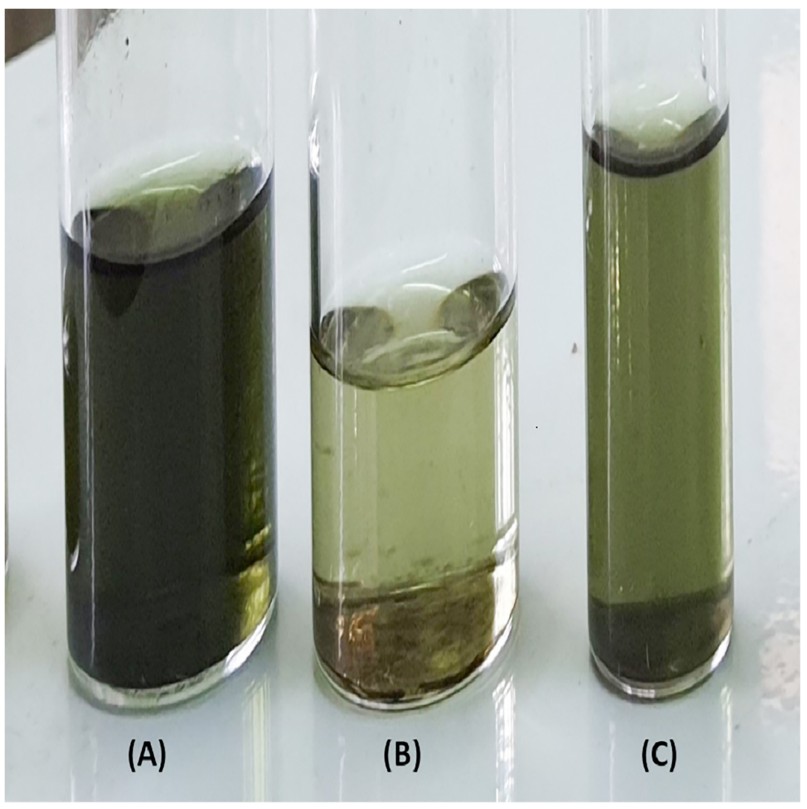

**Figure 8** **Extracted lipid samples by CPAA (A), CPHA (B), and CPTFA (C).**

respectively. This indicates that utilization of the last two CPILs causes pigments'
deterioration, particularly CPHA—which produces a yellow extract, a beneficial outcome
that would potentially minimize the number of steps required in processing biodiesel.
In general, the results reveal that the three CPILs can efficiently extract lipids from
*S. platensis* microalgae and that CPHA aids in lowering the pigment content in the lipid
samples.

## CONCLUSIONS

Six caprolactam-based ionic liquids (CPILs) were investigated for lipid extraction from wet
and dried *S. platensis* microalgae, using both pure CPILs and co-solvent mixtures (CPILs/
methanol), and compared to the conventional organic solvent (methanol/hexane)
extraction method. The pure forms and IL/methanol mixtures of three of these CPILs—
Caprolactamium acetate (CPAA), Caprolactamium chloride (CPHA), and caprolactam
trifluoromethane acetate (CPTFA)—showed higher or similar lipid recovery efficiency
from dry biomass compared to the conventional organic solvent (hexane–methanol)
extraction method. The use of CPAA provided a maximum lipid recovery of 14% and 8%
from dry and wet biomass, respectively. On other hand, CPHA and CPTFA minimized
pigment co-extraction, resulting in reduced purification steps in biodiesel production.
Furthermore, the lipids profiles of the three CPILs were dominated by palmitic acid, oleic

and stearic fatty acids, comparable to those produced by the conventional method. Therefore, the three CPILs are promising green solvents, with potential energy and cost savings in biodiesel production from microalgae. Further studies should investigate the intra molecular interactions of these ILs and their effectiveness for extraction of lipids.

## ACKNOWLEDGEMENTS

The authors would like to acknowledge Moi University for providing access to research facilities.

### Funding
This work was supported by the Africa Centre of Excellence II in Phytochemicals Textile and Renewable Energy (ACE II PTRE). The funders had no role in study design, data collection and analysis, decision to publish, or preparation of the manuscript.

### Grant Disclosures
The following grant information was disclosed by the authors:
Africa Centre of Excellence II in Phytochemicals Textile and Renewable Energy (ACE II PTRE).

### Competing Interests
The authors declare that they have no competing interests.

### Author Contributions

- Rania A. Naiyl conceived and designed the experiments, performed the experiments, analyzed the data, performed the computation work, prepared figures and/or tables, authored or reviewed drafts of the paper, and approved the final draft.
- Fredrick O. Kengara conceived and designed the experiments, prepared figures and/or tables, authored or reviewed drafts of the paper, and approved the final draft.
- Kirimi H. Kiriamiti conceived and designed the experiments, authored or reviewed drafts of the paper, and approved the final draft.
- Yousif A. Ragab conceived and designed the experiments, authored or reviewed drafts of the paper, and approved the final draft.

### Data Availability
The raw data is available in the Supplemental Files.

### Supplemental Information
Supplemental information for this article can be found online at http://dx.doi.org/10.7717/peerj-achem.13#supplemental-information.

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
