# Peer review of "Lipid extraction from microalgae using pure caprolactam-based ionic liquids and with organic co-solvent"

_PeerJ Analytical Chemistry, doi:10.7717/peerj-achem.13_

## Round 0.1 · original submission · Major Revisions

Please carefully address the reviewers' comments.

Reviewer 1 ·

Basic reporting

The authors here describe a protocol of extracting lipids from wet and dried Spirulina platensis microalgae using ionic liquids. This is indeed an interesting work towards a step to follow green methodology for finding resources of biofuels. The manuscript may be published in the PeerJ Analytical Chemistry after considering the points mentioned in the detailed report.

Experimental design

In some cases, the authors have mentioned the experimental procedure to be found in literatures, such as, 'Soxhlet extraction'. Authors may consider to provide a brief description of all such methods to help the readers to immediately have an idea while going through the paper.

Validity of the findings

The results are interesting which are validated by reproducible data set.

Additional comments

1. The authors may clarify if the ionic liquids (ILs) used in the study are the room temperature ILs. Mentioning their melting point is important as a reader may find different uses of these molecules.

2. All the ILs have same cation but differing in their anions. The different effectiveness of extracting lipids seems to depend on the type of anions. Why so?

3. The authors primarily have described their extracted lipids to be used for biofuel. Are they suitable for preparing model membrane, such as, vesicles? A discussion may be added to extend the uses of the lipids.

Reviewer 2 ·

Basic reporting

Species names should be properly mentioned (following the guidelines and put in italic, etc).
Furthermore, I suggest the authors not to overgeneralize the statement from literature and I invite the authors to perform critical reflections on the results.

Experimental design

The water content was unfortunatley not reported, thus the different ionic liquids are not directly comparable.
I especially disagree with the method to determine the yield, which I address several times in the annotated PDF. This could result in misleading interpretation, serious claims and wrong conclusion.

Validity of the findings

Due to the improper method (described above), the authors obtained peculiar data, which seems to be 'too good to be true'. A critical reflection on the obtained results is not observed from the report.
Replicates or standard deviation in some sections were not reported.

Annotated reviews are not available for download in order to protect the identity of reviewers who chose to remain anonymous.

---

## Round 0.2 · accepted · Accept

The authors have addressed the reviewers' comments.

Reviewer 1 ·

Basic reporting

Done in first review

Experimental design

Good

Validity of the findings

During the first review, there were multiple comments and suggested correction. The revised manuscript, the authors has taken care of all of them. Now the manuscript is ready to publish.

Additional comments

During the first review, there were multiple comments and suggested correction. The revised manuscript, the authors has taken care of all of them. Now the manuscript is ready to publish.